# Distributed Temperature Sensing through Network Analysis Frequency-Domain Reflectometry

**DOI:** 10.3390/s24072378

**Published:** 2024-04-08

**Authors:** Rizwan Zahoor, Raffaele Vallifuoco, Luigi Zeni, Aldo Minardo

**Affiliations:** Department of Engineering, Università della Campania Luigi Vanvitelli, Via Roma 29, 81031 Aversa, Italy; rizwan.zahoor@unicampania.it (R.Z.); raffaele.vallifuoco@unicampania.it (R.V.); luigi.zeni@unicampania.it (L.Z.)

**Keywords:** distributed temperature sensing, rayleigh scattering, optical frequency-domain reflectometry

## Abstract

In this paper, we propose and demonstrate a network analysis optical frequency domain reflectometer (NA-OFDR) for distributed temperature measurements at high spatial (down to ≈3 cm) and temperature resolution. The system makes use of a frequency-stepped, continuous-wave (cw) laser whose output light is modulated using a vector network analyzer. The latter is also used to demodulate the amplitude of the beat signal formed by coherently mixing the Rayleigh backscattered light with a local oscillator. The system is capable of attaining high measurand resolution (≈50 mK at 3-cm spatial resolution) thanks to the high sensitivity of coherent Rayleigh scattering to temperature. Furthermore, unlike the conventional optical-frequency domain reflectometry (OFDR), the proposed system does not rely on the use of a tunable laser and therefore is less prone to limitations related to the laser coherence or sweep nonlinearity. Two configurations are analyzed, both numerically and experimentally, based on either a double-sideband or single-sideband modulated probe light. The results confirm the validity of the proposed approach.

## 1. Introduction

Distributed optical-fiber sensors can monitor the continuous spatial profile of temperature and strain, usually exploiting some scattering mechanism occurring in an optical fiber [1]. Distributed sensors based on Brillouin scattering are capable of long-range sensing [2] and, in some cases, high spatial resolution [3,4,5] measurements. However, the measurand resolution is somewhat limited by the relatively low sensitivity of the Brillouin frequency shift to temperature (≈1 MHz/°C) and strain (≈50 kHz/µε). Instead, distributed sensors with functions based on Rayleigh scattering benefit from a much greater sensitivity of the Rayleigh spectrum to these measurands (≈1.3 GHz/°C and 150 MHz/με, respectively) [6]. In order to measure the Rayleigh spectral shift, two approaches are commonly adopted: the optical frequency-domain reflectometry (OFDR), which operates in the frequency domain, and the phase-sensitive optical time-domain reflectometry (ϕ-OTDR), which operates in the time domain. In the former approach, the backscatter Rayleigh complex field is acquired by scanning the frequency of a tunable laser over a wide range while the positional information is recovered through Fourier transform [6]; in the ϕ-OTDR approach, instead, the Rayleigh backscatter is acquired using a pulsed source while the laser frequency is scanned over a relatively narrow range [7]. In both cases, the temperature (or strain) variations are recovered by cross-correlating the Rayleigh spectra acquired in two consecutive measurements. Comparing these two approaches, the time-domain configuration is less prone to limitations associated with laser coherence; in fact, a km-scale sensing range is easily reached. Furthermore, the scan of the laser frequency can be avoided by using linearly chirped probe pulses, which enable faster, single-shot measurements [8]. Nonetheless, the use of a pulsed source usually involves a tradeoff between the signal-to-noise ratio (SNR) and the spatial resolution: using a shorter pulse improves the spatial resolution at the cost of a lower backscattered energy [9]. On the other hand, the OFDR approach ensures a finer spatial resolution owing to the much larger amount of energy injected into the fiber. As a matter of fact, commercial OFDR equipment can reportedly perform distributed temperature measurements with a temperature resolution of ±0.1 °C and a spatial resolution of 1 cm [10].

Recently, we have demonstrated a novel OFDR method based on a single-frequency laser and a vector network analyzer (VNA) [11]. In this technique, named NA-OFDR, the VNA output is employed to impress a small sinusoidal amplitude modulation to the laser light. At the same time, the backscatter signal is mixed with a local oscillator and demodulated by the VNA. The acquired data are subjected to an inverse Fourier transform to recover the spatial information at a spatial resolution determined by the range of modulation frequencies scanned by the VNA [12]. In Ref. [11], the NA-OFDR method was applied to detect vibrations (strain) at a rate much higher than that commonly allowed by OFDR. In this paper, the same NA-OFDR configuration is employed for static temperature measurements. Compared to the conventional OFDR method, the NA-OFDR attains high spatial resolution; additionally, it requires neither a tunable laser, which usually limits the coherence length, nor a trigger interferometer to correct laser-tuning errors [13].

Compared to the NA-OFDR configuration recently applied to vibration sensing, the method proposed here has two fundamental differences: first, we use the amplitude rather than the phase of the Rayleigh backscattered light. Second, the coherent Rayleigh trace is acquired for a discrete set of laser frequencies, which permits reconstruction of the Rayleigh spectrum at each fiber position. The temperature information is then recovered by cross-correlating the Rayleigh spectra acquired in two consecutive measurements, as in conventional OFDR sensors.

As in Ref. [11], two modulation formats of the probe light are analyzed here: single-sideband (SSB) and double-sideband (DSB) modulation. In the following sections, the two configurations are analyzed first numerically and then assessed experimentally, with the results demonstrating their ability to obtain distributed measurements with high spatial and temperature resolution.

## 2. Numerical Analysis

The model developed in Ref. [11] describes the baseband transfer function of the fiber, which was obtained by coherently mixing the Rayleigh backscatter excited by a DSB or an SSB-modulated probe with a local oscillator and demodulated by the VNA. The model calculates the Rayleigh backscatter assuming a random distribution of the scatterers along the fiber. For an optical fiber with N scatterers, the VNA output using a DSB- or SSB-modulated probe is given by [11]:(1)$${S21}_{DSB}\left(\omega_{m}\right)\propto \sum_{i=1}^{N}4a_{i}cos\left(2k_{0}z_{i}\right)e^{j\left(-2\frac{\omega_{m}}{v}z_{i}\right)}.$$
(2)$${S21}_{SSB}\left(\omega_{m}\right)\propto \sum_{i=1}^{N}2a_{i}e^{\left(2jk_{0}z_{i}\right)}e^{j\left(-2\frac{\omega_{m}}{v}z_{i}\right)}.$$

In Equations (1) and (2), $z_{i}$ and $a_{i}$ are the positions and reflection coefficients of the N scatterers; $k_{0}=\frac{\omega_{0}}{v}$ is the wavenumber corresponding to the laser angular frequency $\omega_{0}$; $v=\frac{c}{n}$ is the phase velocity; and $\omega_{m}$ is the angular frequency modulation impressed by the VNA. Given the distribution of the scatterers, Equations (1) and (2) can be used to calculate $S_{21}\left(\omega_{m}\right)$ for a discrete set of modulation frequencies $\omega_{m}$. Then, the spatial distribution of the Rayleigh scatter amplitude (and phase) is obtained by applying an inverse Fourier transform to $S_{21}\left(\omega_{m}\right)$ and converting the time coordinate into a spatial one using $z=\frac{v_{g}}{2}t$, $v_{g}$ being the group velocity of the optical guided mode. The choice of modulation frequencies $\omega_{m}$ is related to the fiber length and target spatial resolution, according to the following relationships [12]:(3)$$SR=\pi \frac{v_{g}}{\omega_{m,max}-\omega_{m,min}}$$
(4)$$L_{max}=\pi \frac{v_{g}}{{∆\omega }_{m}}$$
where SR is the spatial resolution; $L_{max}$ is the maximum length of the fiber; $\omega_{m,max}$ and $\omega_{m,min}$ are the maximum and minimum modulation angular frequencies, respectively; and ${∆\omega }_{m}$ is the angular modulation frequency step.

As the temperature (or strain) change is obtained from the shift of the Rayleigh backscatter power in the laser-frequency domain, the coherent Rayleigh trace must be computed over a range of laser frequencies [7]. When some temperature or strain perturbation acts on the fiber, the positions $z_{i}$ of the scatterers falling within the perturbed area are progressively shifted by an amount that depends on the applied perturbation; all scatterers after the perturbed area are shifted by the same amount. Furthermore, the refractive index in the perturbed area changes due to thermo-optic or elasto-optic effects [14]. The new distribution of the scatterers can then be used to recalculate the fiber response over the same set of laser frequencies. Finally, as the Rayleigh spectrum has a speckle-like appearance, the temperature (or strain) induced shift is determined, at each fiber position, by cross-correlating the Rayleigh spectrum with the one measured under known environmental conditions, as in conventional OFDR sensors.

First, we analyzed the fiber response in absence of perturbations by calculating the autocorrelation of the Rayleigh spectrum. In the following analysis, we assume a fiber length of 60 m and an average distance between the scatterers equal to 1 mm. SR was set to 1 m or 10 cm by calculating $S_{21}\left(\omega_{m}\right)$ up to $\omega_{m,max}=2\pi \times 100\;\mathrm{MHz}$ or 2π×1 GHz, respectively. In both cases, the average distance between the scatterers was much shorter than SR, thus meeting the requirements discussed in Ref. [14]. The laser frequency was varied over an interval of 10 GHz with a frequency step of 25 MHz. The results relative to those from a DSB- or SSB-modulated probe are visualized in Figure 1. The simulation data indicate that, for an SSB modulated-probe, the 3-dB bandwidth of the autocorrelation increases from ≈61 MHz at SR = 1 m to ≈339 MHz at SR = 10 cm. For a DSB-modulated probe, the autocorrelation behaves similarly, with the 3-dB bandwidth increasing from ≈75 MHz at SR = 1 m to ≈297 MHz at SR = 10 cm. The broadening of the correlation peak, similar to the one observed in time-domain schemes [15], is explained by considering that a smaller SR implies a smaller distance between the most distant scatterers inside each resolution cell. Consequently, the interference fringes in the Rayleigh spectrum spread out, with a consequent broadening of the correlation curve. Therefore, setting a smaller SR will inevitably restrict the smallest detectable change in temperature (or strain) due to the resulting increase in the Rayleigh spectrum bandwidth.

More simulations were performed by varying the temperature from 0 °C to 0.2 °C along a 2-m central region of the fiber. For these simulations, the spatial resolution was set to 1 m. We show, in Figure 2, the cross-correlation of the Rayleigh spectra at the center of the hotspot, as computed after the spectra were scaled such that their autocorrelations at zero lag equal unity. As expected, the correlation peak shifts to the left by an amount proportional to the temperature variation. In addition, the simulations reveal that the amplitude of the correlation peak oscillates with the applied perturbation in the case of a DSB-modulated probe, but remained more uniform in the case of an SSB-modulated probe (see Figure 2c). Specifically, the value of such oscillations was as large ≈ 49% for DSB measurements and as large as ≈ 12% for SSB measurements. Still, we attribute this behavior to the interference of the Rayleigh traces produced by the two probe sidebands [13]. Obviously, a too-weak correlation peak could prevent a correct estimate of the Rayleigh frequency shift whenever its amplitude falls below the level of the background noise or the sidelobes. Therefore, according to our analysis, the SSB modulation scheme should be preferred to the DSB modulation scheme to ensure reliable measurements.

## 3. Experimental Results

Figure 3 shows the experimental setup used for measuring the coherent Rayleigh traces based on the NA-OFDR approach proposed here. The setup is essentially identical to the one employed in Ref. [11] for distributed vibration measurements, with the only addition of a variable frequency shifter composed of an SSB electro-optic modulator driven by an RF source, which is placed immediately after the laser. As already explained, the variable frequency shifter is required to scan the laser frequency and therefore acquire the Rayleigh backscatter spectrum. In brief, the light from a 1550-nm narrowband laser is first sent to the frequency shifter. The frequency-shifted light is then split by a 90/10 optical coupler to separate the probe light (upper branch) from the local oscillator (lower branch). The former is first intensity-modulated by an electro-optic modulator (EOM1) biased at its null point and driven by the output port of the VNA; it is then amplified by an erbium-doped fiber amplifier (EDFA). In the DSB-modulated probe configuration, the amplified probe is directly launched into the sensing fiber. By contrast, in the case of SSB modulation, a narrowband (≈8 GHz FWHM) fiber Bragg grating (FBG) with a thermal package (from TeraXion Inc, Québec, QC, Canada), is inserted before the sensing fiber with the aim of selecting only one of the two sidebands generated by EOM1. Note that for SSB modulation tests, an offset of 8 GHz is applied to the modulation frequencies scanned by the VNA, ensuring that the two sidebands produced by EOM1 are separated by at least 2 × 8 = 16 GHz. In the receiver path, the backscattered light is mixed with the local oscillator and the beat signal is detected through a photodetector connected to the input port of the VNA. Note that in SSB-modulated probe experiments, the local oscillator is frequency upshifted through another electro-optic modulator (EOM2) driven by an RF synthesizer at a frequency of 8 GHz. This arrangement makes it possible to balance the offset applied to the VNA modulation frequency sweep. The same offset is also applied to the photodetector output through an RF mixer, upshifting the frequency of the beat signal by 8 GHz and thus ensuring that the VNA input sweeps the same range as its output [11]. In both cases, the mixed light is sent to a 4-GHz photodetector. For SSB experiments, the presence of EOM2 and the RF mixer in the receiver path resulted in an additional insertion loss of 8 dB compared to the results of DSB experiments. Finally, we observe that while both the probe light and local oscillator are frequency-swept in the conventional OFDR, in our NA-OFDR scheme, only the frequency of the probe light is swept, while the local oscillator has a fixed frequency, a point of similarity to the time-gated digital OFDR method [16].

For our experiments, a ≈55-m single-mode fiber (the AcoustiSens^®^ by OFS, Norcross, GA, USA) with an acrylate coating of 200 µm was employed as the fiber under test (FUT). This fiber provides 13 dB more Rayleigh scattering power compared to conventional single-mode fibers, while ensuring a relatively low attenuation loss (≈0.7 dB/km). Obviously, a higher scattering power results in an improved optical signal-to-noise ratio (OSNR) for any given interrogation scheme. We also observe that, although the fiber has a weak grating inscribed along its length (covering the wavelength range 1536–1566 nm), its behavior in terms of Rayleigh backscatter is still well described by the conventional 1-D model used for simulations [11]. For each test, the state of polarization of the local oscillator was adjusted to maximize the average power of the detected signal. In DSB-modulated probe experiments, the laser frequency was varied by 10 GHz with a step of 25 MHz by means of the frequency shifter. In SSB-modulated probe experiments, however, the laser frequency was swept by 4 GHz only. The reason for using this reduced sweep stems from the necessity, in case of SSB-modulated probe experiments, of keeping the selected probe sideband within the reflection bandwidth of the FBG while varying both the laser and the VNA modulation frequency. In the case of DSB experiments, no FBG was employed; therefore, the laser frequency range was limited only by the bandwidth of the frequency shifter. All tests were carried out with an intermediate frequency (IF) bandwidth of the VNA set equal to 100 kHz.

The first experiments were conducted by sweeping the VNA modulation frequency from 1 MHz to 100 MHz with a step of 1 MHz. These settings resulted in SR = 1 m. A hotspot was created by immersing a 3-m central piece of the FUT in a water bath, whose temperature was monitored through a thermocouple. Note that the fiber was immersed directly, without a protective tube. As an example, we show in Figure 4a,b the cross-correlation of the Rayleigh spectra acquired for a water-temperature variation of 0.1 °C, corresponding to a Rayleigh frequency shift of ≈−130 MHz. The same graphs also show the retrieved Rayleigh frequency shift profile. Repeating the measurement for several water-bath temperatures at a step of 0.1 °C, a calibration factor of −1.33 GHz/°C was derived (see Figure 4c). Using this factor, the average temperature variation in the hotspot region shown in Figure 4a,b was found to be ≈0.14 °C for the DSB measurement and ≈0.13 °C for the SSB measurement. The discrepancy between the nominal and measured temperature changes is mostly attributed to the limited accuracy with which the temperature in the water bath could be set (≈0.1 °C). Besides, an additional uncertainty arises from the swelling of the acrylate coating as a result of water absorption [17]. Note that in both the DSB and SSB experiments, the absolute Rayliegh frequency shift (≈130 MHz) was much less than the laser frequency sweep, thus ensuring that the result of the cross-correlation was correct [18].

In Figure 5, we show the temperature-change profiles acquired using either a DSB- or SSB- modulated probe for the whole set of water-bath temperatures. We note that the DSB-modulated probe experiments were conducted by varying the temperature of the water bath up to ≈ 2.9 °C, versus ≈ 1.1 °C in the SSB-modulated probe experiments. Such a difference derives from the different range swept by the laser frequency in the two configurations (10 GHz for the DSB probe, 4 GHz for the SSB probe). We also observe that the hotspot size in the reconstructed profiles is progressively reduced when the temperature variation increases. This change is attributed to errors in the frequency shift estimation near the transition regions caused by the finite spatial resolution [15]. The average standard deviation of the temperature profiles outside the hotspot was ≈7 mK in DSB measurements and ≈11 mK in SSB measurements. The difference between the temperature uncertainties of the two schemes is attributable to the lower SNR of SSB measurements, as already discussed. In both figures, the inset shows the amplitude of the correlation peak at the hotspot position. It is seen that in both cases, the correlation peak decays when the temperature variation increases, an obvious consequence of the limited range of the scanned laser frequencies. We also notice that the correlation peak amplitude for the DSB probe experiments does not exhibit the strong oscillations predicted by our numerical analysis. This finding can be attributed to the very short period of these oscillations, which are difficult to observe experimentally. For example, the simulation data in Figure 2 indicate that these oscillations have a period of only 25 MHz, which corresponds to a temperature period of less than 20 mK. Under practical conditions, it is difficult to control the temperature with such accuracy. In other words, the fluctuations in the water-bath temperature over time, occurring during the acquisition of the Rayleigh traces, may induce a “smearing out” of the measurement data compared to the results of the simulation.

Further experiments were carried by scanning the VNA modulation frequency up to 3 GHz while keeping the frequency step equal to 1 MHz. This change resulted in SR ≈ 3 cm. For these tests, two 5-cm pieces of the same FUT, located ≈1.7 away from each other, were fixed through silicone thermal grease to a Peltier cell connected to a PID controller. The fiber was only loosely fixed with the thermal paste so as to avoid any strain transfer to the fiber. A 1.2-mm thermistor-bead was also fixed to the border of the Peltier cell using the same thermal grease. The thermoelectric system formed by the Peltier cell, the thermistor and the PID controller allowed us to control the temperature of the fiber pieces fixed to the Peltier cell with a precision better than 0.1 °C. As in the previous tests, the laser frequency was scanned by 10 GHz in DSB experiments or 4 GHz in SSB experiments, while the frequency step was set to 50 MHz. As an example, we show in Figure 6 the cross-correlation of the Rayleigh spectra acquired for a nominal temperature variation of the hotspots equal to 1 °C, with the retrieved Rayleigh frequency shift profiles superimposed. In both cases, the two hotspots are detected by the sensor, although the noise level is apparently higher than what is seen in Figure 4.

Finally, we show in Figure 7 the temperature profiles acquired for nominal changes of temperature at the two hotspot positions equal to 0.5 °C, 1 °C, and 1.5 °C, respectively.

We notice that while both configurations detect the two hotspots, the SSB measurements are slightly noisier than the DSB measurements: specifically, the standard deviation of the temperature profiles measured outside the perturbed zones (averaged over the three tests) was ≈51 mK for the DSB measurements and ≈55 mK for the SSB measurements. Still, we attribute this difference to the lower SNR of SSB measurements. As regards the temperature at the hotspots, the maximum deviation between the measured and nominal temperature was 0.28 °C for the DSB measurements and 0.26 °C for the SSB measurements. We mostly attribute these errors to the difficulty of controlling the temperature of the fiber at the hotspots. In fact, even though the Peltier cell could control the temperature with a precision better than 0.1 °C, only a portion of the fiber was in contact with the cell, leaving the rest exposed to air. Possible residual strain due to the thermal expansion of the thermal grease could also contribute to these errors.

While the results in Figure 7 show that DSB and SSB modulation formats provide similar performance, we must recall that these results were obtained with two different laser frequency sweeps. Interestingly, when the DSB measurements were truncated to the same laser frequency sweep adopted in SSB measurements (4 GHz), the standard deviation of the temperature profiles outside the hotspots increased to 81 mK, which was worse than that of the SSB measurements despite the 8-dB SNR advantage. Thus, it may be argued that if a fair comparison between the two configurations could be made (i.e., ensuring the same SNR and the same laser frequency sweep range), the SSB configuration should be expected to provide more accurate results than the DSB configuration, as in the results presented in Ref. [11].

## 4. Conclusions

In this paper, the performance of the NA-OFDR system was demonstrated for distributed temperature (and strain) measurements. The system is capable of obtaining high spatial and temperature resolution; therefore, it can find application whenever small, localized temperature changes must be detected. While the experiments were performed by applying small variations in temperature, larger variations could be accommodated by setting the reference to the measurement immediately preceding each new measurement and summing incremental spectral shifts across all times [8,19].

Unlike the conventional OFDR method, the NA-OFDR method does not rely on a tunable laser and is therefore less prone to the limitations imposed by the phase noise and nonlinearity of the frequency sweep [20]. Furthermore, the use of a VNA for signal demodulation is compatible with high-resolution Brillouin sensing [5], which makes this technique ready to be extended to hybrid multiparameter sensing [21,22]. Finally, the technique can be easily combined with the method demonstrated in Ref. [11] for vibration measurements, making it well suited for simultaneous static and dynamic measurements.

One major issue is the frequency drift of the laser source, which causes a progressive degradation of the cross-correlation quality. In our measurements, we observed a significant drop in the correlation peak amplitude when correlating a reference trace with a measurement taken after several minutes (during which the frequency drift of our laser module should be a few MHz in magnitude, according to the manufacturer). In a future analysis, we plan to improve the long-term stability of our system by locking the laser frequency to the absorption line of a gas molecule, as in Ref. [7]. Furthermore, we plan to conduct SSB experiments using an optical I/Q receiver, which would allow the separation of the Rayleigh backscatter contributions from the two sidebands without incurring additional losses.

## Figures and Tables

**Figure 1 sensors-24-02378-f001:**
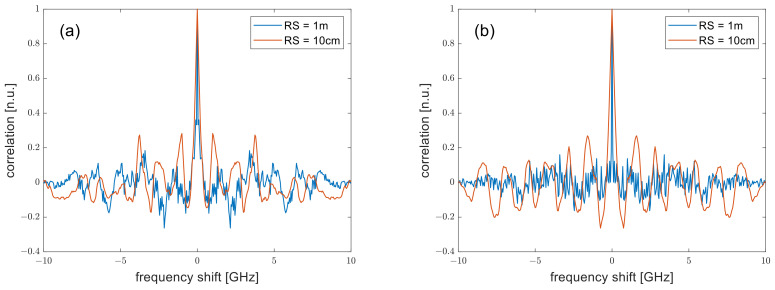
Autocorrelation of the Rayleigh spectrum as computed for (**a**) a DSB-, or (**b**) an SSB-modulated probe, for SR = 1 m or SR = 10 cm.

**Figure 2 sensors-24-02378-f002:**
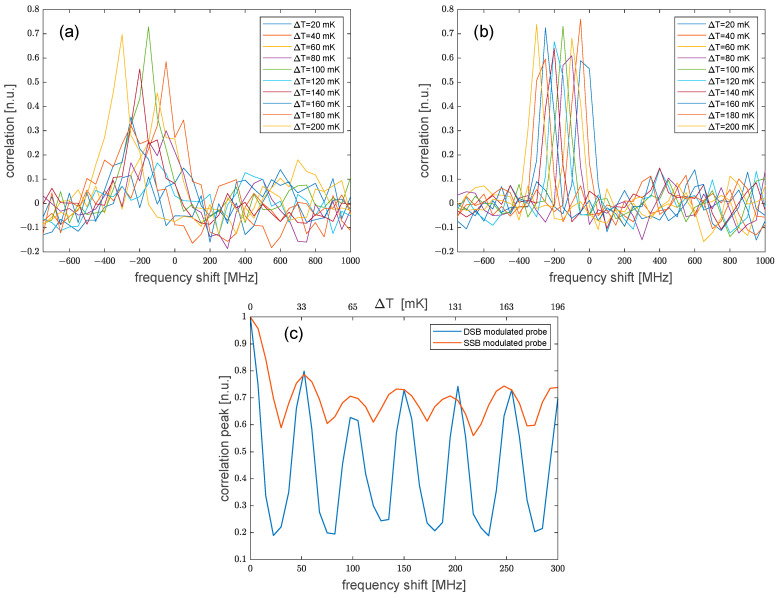
Cross-correlation of the Rayleigh spectrum as computed for (**a**) a DSB-, or (**b**) an SSB-modulated probe; (**c**) Amplitude of the correlation peak versus the perturbation-induced spectral shift.

**Figure 3 sensors-24-02378-f003:**
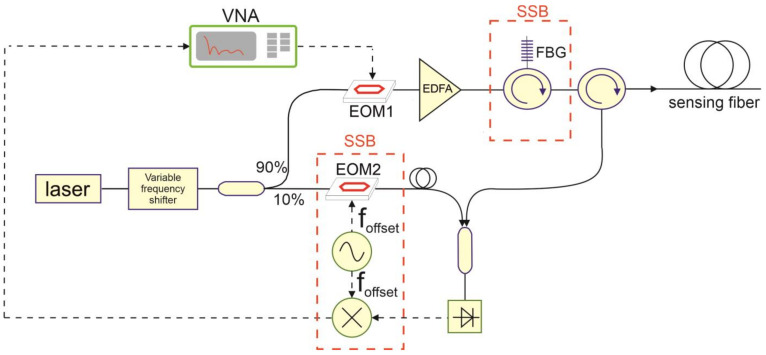
Experimental setup for NA-OFDR temperature measurements. The components inside the red dashed box are used only for SSB-modulated probe measurements. EDFA: Erbium-doped fiber amplifier; FBG: fiber Bragg grating; EOM: electro-optic modulator; VNA: vector network analyzer.

**Figure 4 sensors-24-02378-f004:**
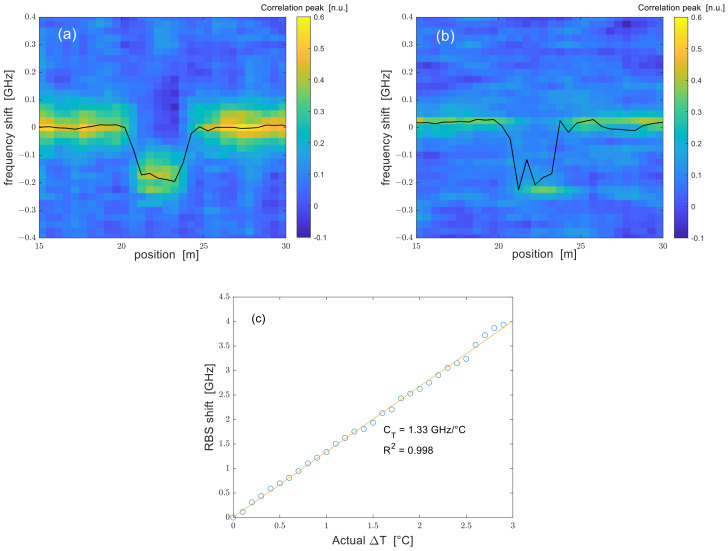
Cross-correlation of the Rayleigh spectra acquired at SR = 1 m using a (**a**) DSB- or (**b**) SSB-modulated probe with a nominal temperature variation of 0.1 °C from z = 21 m to z = 24 m. The superimposed black lines represent the Rayleigh frequency shift profile; (**c**) Rayleigh backscatter shift as a function of the applied temperature change.

**Figure 5 sensors-24-02378-f005:**
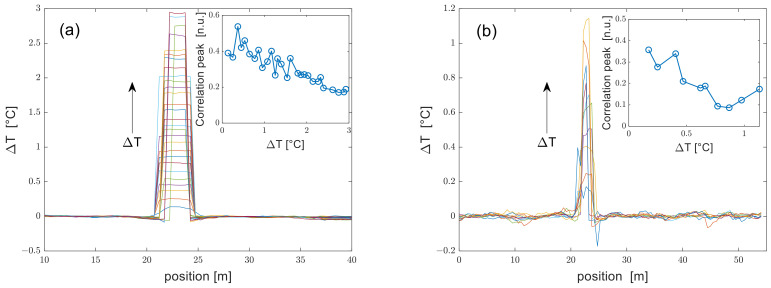
Temperature profiles acquired using (**a**) a DSB- or (**b**) an SSB-modulated probe at SR = 1 m. The insets show the correlation peak amplitude at the perturbated position at varying applied temperature changes.

**Figure 6 sensors-24-02378-f006:**
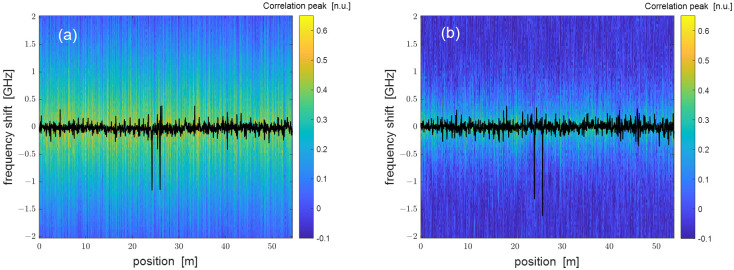
Cross-correlation of the Rayleigh spectra acquired using a (**a**) DSB- or (**b**) SSB-modulated probe with a nominal temperature variation of 1 °C along two 5-cm spots located at z ≈ 24.1 m and z ≈ 25.8 m and SR ≈ 3 cm. The superimposed black lines represent the Rayleigh frequency shift profile.

**Figure 7 sensors-24-02378-f007:**
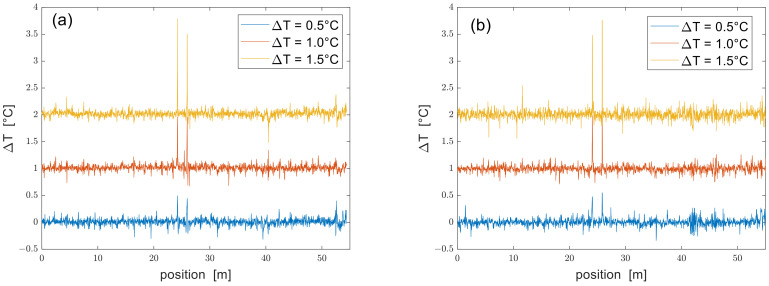
Temperature profiles acquired using (**a**) a DSB- or (**b**) an SSB-modulated probe at SR ≈ 3 cm. The profiles are vertically shifted by 1 °C each other for clarity purposes.

## Data Availability

Data are available upon reasonable request.

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
