# Peer review of "Distributed Temperature Sensing through Network Analysis Frequency-Domain Reflectometry"

_sensors, 2024, doi:10.3390/s24072378_

Round 1
Reviewer 1 Report
Comments and Suggestions for Authors
The authors propose and demonstrate a network analysis optical frequency domain reflectometer (NA-OFDR) for distributed temperature measurements at high spatial (down to ≈ 38cm) and temperature resolution. Some issues must be clarified before the manuscript is accepted for publication in Sensors.
1. What is the unit “mk” in the manuscript? Is it mkHz?
2. In Figure 3, what is the function of the frequency shifter? The laser is a single frequency laser, when the frequency of the laser has been shifted, it is just another single frequency laser.
3. The bandwidth of the spectrum reflected from FBG used in Figure 3 is about 8GHz which corresponds to 0.06nm, and the reflected wavelength of FBG will drift when the ambiance temperature varies. The temperature sensitivity of the FBG is about 13pm/°C, when the temperature of FBG varies, how to ensure that the FBG will reflect the light?
4. The fonts of the title horizontal coordinates and vertical coordinates of the figures are too small and it is difficult to read them clearly.
Author Response
The authors propose and demonstrate a network analysis optical frequency domain reflectometer (NA-OFDR) for distributed temperature measurements at high spatial (down to ≈ 38cm) and temperature resolution. Some issues must be clarified before the manuscript is accepted for publication in Sensors.
- What is the unit “mk” in the manuscript? Is it mkHz?
The unit mK stands for milliKelvin. As we refer to temperature changes, in our context mK coincides with m°C (milliCelsius). We adopted mK instead of m°C for brevity.
- In Figure 3, what is the function of the frequency shifter? The laser is a single frequency laser, when the frequency of the laser has been shifted, it is just another single frequency laser.
We understand the perplexity of the Reviewer. We have replaced the text “frequency shifter” in Figure 3, with the text “variable frequency shifter”. The purpose of this variable frequency shifter is to sweep the laser frequency over a proper range, and thus acquire the Rayleigh backscatter spectrum. In fact, the adopted sensing method recovers the temperature (or strain) variations, based on the shift of the Rayleigh spectrum, with the latter being reconstructed by acquiring Rayliegh backscatter trace over a range of laser frequencies. In our method, the laser frequency was shifted by means of an SSB modulator driven by a RF source. Alternatively, a tunable laser could be used for this same purpose. However, tunable lasers have limited coherence length and reduced linearity. For this reason, we have adopted a solution based on a laser at fixed frequency, followed by a variable frequency shifter. A sentence about the choice of a frequency shifter instead of a tunable laser, and a new reference (Ref. [13]), have been added to page 2.
- The bandwidth of the spectrum reflected from FBG used in Figure 3 is about 8GHz which corresponds to 0.06nm, and the reflected wavelength of FBG will drift when the ambiance temperature varies. The temperature sensitivity of the FBG is about 13pm/°C, when the temperature of FBG varies, how to ensure that the FBG will reflect the light?
As correctly noted by the Reviewer, a large temperature-induced shift of the FBG reflection spectrum may lead to a malfunctioning of our setup. To avoid this issue, we have utilized an FBG with athermal packaging (by Teraxion Inc), reducing the thermal drift of the Bragg grating to less than 0.5 pm/°C (https://www.teraxion.com/en/blog/narrow-optical-filter-wavelength-stabilization/). This point has been clarified in the text.
- The fonts of the title horizontal coordinates and vertical coordinates of the figures are too small, and it is difficult to read them clearly.
The font size of all figures has been increased to improve readability.
Reviewer 2 Report
Comments and Suggestions for Authors
In the PDF you will find some details for specific comments /remarks.
The following comments should be applied to the whole article:
Figure:
The font size of the figures needs to be increased.
Most figures need a legend for clarification.
Figures 4 & 6 need a colour bar to validate e.g. SNR.
Please change the scaling of the images to the regions of interest.

Author Response
In the PDF you will find some details for specific comments /remarks.
We have made some revisions based on the Reviewer’s comments/remarks reported in the PDF. In detail:
1) The advantage of avoiding the use of a tunable laser and trigger interferometer has been clarified, also adding a new reference (Ref. [13]).
2) We have added details about the AcoustiSens fiber (operational wavelength range, coating diameter, advantage compared to conventional fibers).
3) We have added the meaning of the acronym IF.
4) We specify that no protective tube was used before immersing the fiber into the water bath.
5) We have provided the temperature accuracy of the water bath used for experiments.
6) We have clarified the sentence relative to the temperature fluctuations.
7) We have clarified in the manuscript that the fiber was only loosely stuck with thermal paste on the Peltier cell, therefore it was not subject to strain (at least nominally).
8) A new comment has been after Fig. 6.
9) The hotspot positions were specified in the caption of Figs. 4 and 7.
10) A comment about possible residual strain induced by the thermal grease expansion has been added in the revised manuscript
The following comments should be applied to the whole article:
The font size of the figures needs to be increased. Most figures need a legend for clarification.
The font size of all figures has been added, while legends have been added where requested by the Reviewer.
Figures 4 & 6 need a colorbar to validate e.g. SNR.
Color bars were added to Figures 4 and 6.
Please change the scaling of the images to the regions of interest.
The scaling of Fig. 4 has been changed as suggested by the Reviewer.
This point raises the question of why experiments are not carried out in a comparable way.
The reason why the DSB and SSB measurements were carried out using two different frequency sweep ranges is explained in page 6: in SSB-modulated probe experiments, the selected sideband must lie within the FBG reflection bandwidth. In case of DSB experiments, instead, no FBG was employed, therefore the laser frequency range was only limited by the bandwidth of our frequency shifter. For this reason, DSB measurements were carried out with a 10-GHz sweep range, while SSB measurements were carried out with 4 GHz sweep range.
Furthermore, the reason for the 8dB (not mentioned before) should be clarified, comparable to the JLT article of the authors.
As explained in page 5, SSB measurements suffer from an 8-dB extra loss due to the presence of EOM2 and the RF mixer in the receiver path. As a next development, we plan to perform SSB experiments using an I/Q optical receiver, which would allow the discrimination of the two sidebands without incurring in additional loss. A comment on this has been added in the conclusions paragraph.
Reviewer 3 Report
Comments and Suggestions for Authors
In this manuscript, the authors introduce a temperature sensing technique using NA-OFDR, offering high spatial and temperature resolution. Initially, they discuss the problem setup, outlining various temperature sensors employing distributed optical fibers and addressing associated challenges. Subsequently, they demonstrate their technique employing a single-frequency laser with minor modulation from VNA, and conduct analyses to characterize the relationship between peak frequency in the correlation curve and temperature changes. Finally, the authors validate their theory through experiments. Overall, the manuscript is concise and effectively presents the core concept. However, it could benefit from addressing the following comments:
1. In Eqs 2, the exponential function is expressed using two different formats; the authors may want to ensure consistency.
2. The term "spatial resolution" is used frequently after being defined as "SR". Using abbreviations, if defined, would enhance conciseness.
3. Figures 2 and 5 lack immediate clarity regarding the meaning of each curve. It would be beneficial to provide more detailed explanations and descriptions in the figure captions.
4. The font size is too small in most of the figures and make it difficult to read.
5. The temperature sensor appears to heavily rely on the frequency shift in the correlation curve. Although the authors mention that the relationship is 'proportional', providing a calibration curve would offer further insight.
6. The authors suggest that the SSB modulation scheme should be preferred over the DSB modulation scheme for reliable measurements. However, it is surprising that the DSB modulation scheme shows more stable results. Further information regarding the different frequency modulation ranges of the measurements would be helpful.
Author Response
In this manuscript, the authors introduce a temperature sensing technique using NA-OFDR, offering high spatial and temperature resolution. Initially, they discuss the problem setup, outlining various temperature sensors employing distributed optical fibers and addressing associated challenges. Subsequently, they demonstrate their technique employing a single-frequency laser with minor modulation from VNA, and conduct analyses to characterize the relationship between peak frequency in the correlation curve and temperature changes. Finally, the authors validate their theory through experiments. Overall, the manuscript is concise and effectively presents the core concept. However, it could benefit from addressing the following comments:
- In Eqs 2, the exponential function is expressed using two different formats; the authors may want to ensure consistency.
The same format was for the exponential function in Fig. 2 to ensure consistency.
- The term "spatial resolution" is used frequently after being defined as "SR". Using abbreviations, if defined, would enhance conciseness.
After definition, we have replaced the expression “spatial resolution” with SR to enhance conciseness.
- Figures 2 and 5 lack immediate clarity regarding the meaning of each curve. It would be beneficial to provide more detailed explanations and descriptions in the figure captions.
To improve clarity, we have added legends to Fig. 2 and split the results in three subfigures. Furthermore, the caption of Fig. 5 has been revised to give a more detailed explanation.
- The font size is too small in most of the figures and make it difficult to read.
We have increased the font size in all figures.
- The temperature sensor appears to heavily rely on the frequency shift in the correlation curve. Although the authors mention that the relationship is 'proportional', providing a calibration curve would offer further insight.
The calibration curve was added in the revised manuscript as Fig. 4(c).
- The authors suggest that the SSB modulation scheme should be preferred over the DSB modulation scheme for reliable measurements. However, it is surprising that the DSB modulation scheme shows more stable results. Further information regarding the different frequency modulation ranges of the measurements would be helpful.
As correctly noted by the Reviewer, while simulations (and theory) indicate that the SSB modulation scheme is preferable to the DSB modulation scheme, the experimental results show the opposite, with the DSB measurements providing more stable results than SSB measurements. The reasons of this seaming paradox are two: first, DSB measurements were carried out adopting a laser frequency scan of 10 GHz, while for SSB measurements the laser frequency scan was limited to 4 GHz. These data are reported in page 6 and page 8. As discussed in page 10, truncating the DSB measurements to the same range of laser frequency sweep adopted for SSB measurements (4 GHz), the performance of DSB measurements become worse than SSB measurements. The second reason is the 8-dB extra loss due to the components inserted in the local oscillator path (we discuss this in page 5 and page 10). As a future development, we plan to perform SSB measurements using an I/Q optical receiver, which would allow the separation of the scattering from two sidebands, without incurring in additional loss. A comment on this has been added in the conclusions paragraph.
Round 2
Reviewer 1 Report
Comments and Suggestions for Authors
The authors have revised the manuscript with great care, and the manuscript can be accepted for publication in Sensors.
Author Response
We thank the Reviewer for his/her positive comment.
Reviewer 2 Report
Comments and Suggestions for Authors
Open request from last review:
4) Please add the information regarding as mentioned in the authors response.
I. Could the authors gives more details regarding the frequency drift or provide a Allan deviation of the frequency drift?
Please find new comments and remarks in the attached PDF.

Author Response
We thank the Reviewer for his/her new comments/suggestions. We have revised the manuscript accordingly. Please find below our point-to-point reply:
1) Could the authors please discuss the ratio between correlation peak to oscillation (noise)?
We have included the oscillation amplitude for both DSB and SSB measurement data, calculated as the percentage ratio between the standard deviation and the average of the correlation peak in the observed interval.
2) Figure 2. What was temperature difference for the Cross-correlation?
We have modified Figure 2 in order to show, along the upper border, the temperature difference values corresponding to the frequency shift values reported along the bottom border.
3) Please add the information regarding as mentioned in the authors response (4). Furthermore, the influence of water absorption (strain) should be discussed as no protective tube was used.
We have added this piece of information in line 223 and 233 of the revised manuscript, as well as a new reference (Ref. [18]).
4) Figure 5. Please limit the x-axis to a range of 15-30 metres. It is recommended to use the same y-scale in both plots. Please ensure that the insets have the same range on both the x and y axes.
We have changed the x-scale in Fig. 5 and corresponding insets, zooming into the interested area. As regards the y scale, we prefer to leave two different vertical scales for Fig.5a and 5b, in order to have the best visibility for both figures.
5) Figure 6. The authors should add units to the colorbar and ensure that the range is consistent.
We have added units to the colorbar in Fig. 4 and Fig. 6 and ensured consistency of ranges.
6) Could the authors give more details regarding the frequency drift or provide a Allan deviation of the frequency drift?
We didn’t perform any direct measurement of the frequency drift of the laser. However, according to what reported by the manufacturer (Luna RIO), our Orion laser module has a typical frequency drift of ±1.5 MHz for a 30s measurement time, ±4 MHz for a 1-hour measurement time, and ±20 MHz for a 12-hour measurement time. We added this piece of information in the conclusion paragraph.